# External Validation of the Colon Life Nomogram for Predicting 12-Week Mortality in Dutch Metastatic Colorectal Cancer Patients Treated with Trifluridine/Tipiracil in Daily Practice

**DOI:** 10.3390/cancers14205094

**Published:** 2022-10-18

**Authors:** Patricia A. H. Hamers, G. Emerens Wensink, Maarten van Smeden, Geraldine R. Vink, Lidwien P. Smabers, Renee A. Lunenberg, Miangela M. Laclé, Miriam Koopman, Anne M. May, Jeanine M. L. Roodhart

**Affiliations:** 1Department of Medical Oncology, University Medical Center Utrecht, Utrecht University, 3508 GA Utrecht, The Netherlands; 2Department of Epidemiology, Julius Center for Health Sciences and Primary Care, University Medical Center Utrecht, Utrecht University, 3584 CG Utrecht, The Netherlands; 3Department of Research and Development, Netherlands Comprehensive Cancer Organisation, 3511 DT Utrecht, The Netherlands; 4Department of Pathology, University Medical Center Utrecht, Utrecht University, 3508 GA Utrecht, The Netherlands

**Keywords:** trifluridine/tipiracil, metastatic colorectal cancer, Colon Life nomogram, prediction model, daily practice

## Abstract

**Simple Summary:**

Predicting prognosis in cancer patients is needed to guide decision making. In order to predict survival, nomograms can be used to estimate chances of survival based on clinical characteristics. In order to identify metastatic colorectal cancer (mCRC) patients with a very short life expectancy (less than 12 weeks) after receiving multiple standard treatments, the Colon Life nomogram was previously developed. Before a nomogram can be used in daily practice, it is essential to show that it accurately predicts survival in different real-life populations and can be used to guide clinical decision making. This is called external validation. We externally validated the Colon Life nomogram in a cohort of patients with refractory mCRC who were treated with a last treatment option, trifluridine/tipiracil, in daily practice. We demonstrated that the nomogram severely overestimated 12-week mortality and therefore should not be used in clinical practice in its present form. We also showed that quality of life reported by patients themselves can improve the prediction of survival, stressing the importance of patient-reported outcomes. We recommend conducting a study with a sufficiently large sample size to update the Colon Life nomogram or to develop a new model and include quality of life.

**Abstract:**

Background: Predicting prognosis in refractory metastatic colorectal cancer (mCRC) patients is needed to guide decision making. The Colon Life nomogram was developed to predict 12-week mortality in refractory mCRC patients. The aim of this study is to validate the Colon Life nomogram in last line/refractory patients receiving trifluridine/tipiracil (FTD/TPI) in daily practice. Methods: The validation cohort consists of 150 QUALITAS study patients, an observational substudy of the Prospective Dutch CRC cohort, who were treated with FTD/TPI between 2016 and 2019. Model performance was assessed on discrimination, calibration, and clinical usefulness. The additional prognostic value of baseline quality of life (QoL) and thymidine kinase (TK1) expression in tissue was explored. Results: Of the 150 patients, 25 (16.7%) died within 12 weeks of starting FTD/TPI treatment. The C-statistic was 0.63 (95% C.I. 0.56–0.70). The observed/expected ratio was 0.52 (0.37–0.73). The calibration intercept and slope were −1.06 (−1.53 to −0.58) and 0.41 (0.01–0.81), respectively, which indicated overestimation of 12-week mortality by the nomogram. Decision curve analysis showed the nomogram did not yield a positive net benefit at clinically meaningful thresholds for predicted 12-week mortality. Addition of QoL to the nomogram improved the C-statistic to 0.85 (0.81–0.89). TK1 expression was associated with progression-free survival but not with overall survival. Conclusion: We demonstrated evident miscalibration of the Colon Life nomogram upon external validation, which hampers its use in clinical practice. We recommend conducting a study with a sufficiently large sample size to update the Colon Life nomogram or to develop a new model including QoL.

## 1. Introduction

Trifluridine/tipiracil (FTD/TPI) is a treatment available for refractory metastatic colorectal cancer (mCRC) patients [1]. The phase III RECOURSE trial demonstrated a statistically significant but modest survival benefit for FTD/TPI over placebo in pretreated mCRC patients with a median overall survival (OS) of 7.1 months versus 5.3 months, respectively [2]. The median progression-free survival (PFS) after FTD/TPI initiation was 2–3 months in clinical trial and real-world study patients [2,3,4]. The limited median PFS and low response rate (44% [2]) imply that many patients do not benefit from FTD/TPI treatment, resulting in an urgent need for adequate patient selection to avoid unnecessary treatment-related toxicity and to increase cost-effectiveness.

The Colon Life nomogram aims to identify mCRC patients eligible for later treatment lines or inclusion in clinical trials by estimating life expectancy [5]. The suggested approach is that in patients with a predicted limited life expectancy, oncologists should consider withholding further treatment [5]. Pietrantonio et al. [5] developed the nomogram to predict 12-week mortality from diagnosis of refractory mCRC in 411 Italian mCRC patients who received FTD/TPI (*n* = 27, 6.6%), regorafenib (*n* = 113, 27.5%), or other treatments (*n* = 271, 65.9%) between 2006 and 2015. The nomogram includes primary tumour resection, ECOG PS (Eastern Cooperative Oncology Group performance status), lactate dehydrogenase (LDH) value, and peritoneal involvement as predictor variables, with the latter three variables determined upon diagnosis of refractory disease. The authors who developed the nomogram also performed three external validations: in a cohort similar to the development cohort [5], in patients treated with FTD/TPI in the Italian compassionate use program [6], and in RECOURSE trial participants [7]. External validation of the Colon Life nomogram in patients receiving FTD/TPI in daily practice has not yet been described. Based on the nomogram’s discriminatory ability in the external validation studies, the Colon Life nomogram was found to be “an accurate tool for estimating life expectancy and the risk of death” in patients with refractory mCRC [5,6,7]. The calibration of a prediction model, which measures the agreement between predicted and observed outcome risks, is a second important aspect of model performance. Like discrimination, it also influences the clinical usefulness of a prediction model [8,9].

Two parameters not considered in the development of the Colon Life nomogram which have since received attention as possible prognostic markers in mCRC patients receiving FTD/TPI treatment are quality of life (QoL) [3] and thymidine kinase (TK1) expression [10,11]. In a recent study, a low QoL at start of treatment was independently associated with shorter OS and PFS^3^. In addition to QoL, TK1 may also be of prognostic value as it has a role in the metabolism of trifluridine. Trifluridine, the active component of FTD/TPI, is incorporated into tumour DNA, causing DNA damage after phosphorylation by the enzyme TK1 [12]. In patients who were treated with FTD/TPI, high levels of TK1 expression (>10%) was associated with longer OS and PFS [10]. Similarly, patients with high TK1 expression (>15%) demonstrated improved OS when treated with FTD/TPI versus placebo [11]. Therefore, we analysed whether adding QoL and/or TK1 to the Colon Life nomogram increased its prognostic value.

In this study, we aimed to externally validate the Colon Life nomogram in a Dutch real-world mCRC patient population receiving FTD/TPI. Moreover, we explored the additional prognostic value of QoL and TK1 expression to the nomogram.

## 2. Methods

### 2.1. External Validation Cohort

We externally validated the Colon Life nomogram in 150 patients who participated in the QUALITAS study, a substudy of the Prospective Dutch CRC cohort (PLCRC) [3,13]. The QUALITAS study assessed QoL and survival of mCRC patients who were treated with FTD/TPI in daily practice in the Netherlands between 2016 and 2019. Details on study design, inclusion criteria, data collection, baseline characteristics, assumptions, and definitions were published previously [3]. For the current study, the following additional variables, determined after stopping the previous treatment line and prior to starting FTD/TPI treatment, were collected: LDH value, CEA level, leucocyte count, and ECOG PS.

### 2.2. External Validation of the Colon Life Nomogram

#### 2.2.1. Sample Size Calculations

We applied the recommendations published by Riley et al. [14] to determine the minimum sample size recommended for external validation for the Colon Life nomogram. For this, we used the reported C-statistic for the Colon Life nomogram in published external validation studies as the anticipated minimum C-statistic for the Colon Life nomogram and an observed/expected (O/E) ratio of 1.0 to determine the expected confidence interval width within our cohort. We calculated that in our cohort of 150 patients and 25 events, we would be able to detect an O/E ratio of 1.0 with a 95% confidence interval (C.I.) of 0.64–1.36 and a C-statistic of 0.72 with a 95% C.I. of 0.61–0.83, assuming that the linear predictor distributions are normal with a common variance in each outcome group.

#### 2.2.2. Missing Data and Imputation

Certain candidate predictors had missing data. To avoid loss of information and selection bias, we used multiple imputation using Multivariate Imputation by Chained Equations (MICE) [15], assuming missingness at random. The imputation model contained the Colon Life nomogram predictor variables including restricted cubic spline transformations (to accommodate congeniality), the outcome (12-week mortality), and joint model-derived baseline EORTC QLQ-C30 Summary Score (QoL-SS) for QoL, time from diagnosis of mCRC to starting date of FTD/TPI, number of metastatic sites, age, CEA and leukocyte count (with the latter four prior to starting FTD/TPI) as auxiliary variables. We generated 29 imputed datasets, based on the percentage of patients with at least one missing key variable.

#### 2.2.3. Assessing Model Performance

The full multivariable binary logistic regression model, including the coefficients, intercept, and splines were provided by Pietrantonio et al. who developed the nomogram (Appendix A).

Following multiple imputation, 12-week mortality probabilities were estimated for all patients using this model. Performance of the Colon Life nomogram in QUALITAS patients was assessed through discrimination, calibration, and clinical usefulness. Each performance measure was determined for each imputed dataset separately and pooled using Rubin’s rules. Predicted 12-week mortality rates were pooled after complementary log-log transformation.

First, discrimination was determined by the C-statistic. The C-statistic, equivalent to the area under the Receiver Operating Characteristic (ROC) curve, is a measure of how well the model discriminates between those who develop the event (die) and those who do not (survive). The C-statistic is interpreted as the probability that a randomly selected patient who died will have a higher probability of death than a randomly selected subject who survived. Values close to one indicate perfect discrimination ability, while values close to 0.5 indicate poor discrimination ability. Second, calibration was assessed by multiple methods [16,17]. The ratio of observed versus expected number of events (O/E-ratio) was pooled after logarithmic transformation. This ratio should be close to one if the model calibrates well in the validation dataset. Then, model calibration was assessed visually with a calibration plot (a scatter plot of predicted versus observed outcome probabilities) for which well-calibrated predictions will lie along the 45° line [16,17]. Finally, calibration was assessed by fitting a logistic regression model with the observed outcome as the dependent variable, and the log-odds of transformed model predictions as the independent variable. For a perfectly calibrated model, the model intercept and slope would be 0 and 1, respectively. Note that our sample size was insufficient for updating the model in case of miscalibration. As the third step of model performance evaluation, clinical usefulness was assessed by decision curve analysis to determine whether use of a prediction model to inform decision making in clinical practice may result in net benefit for patients [18,19,20]. A model can be recommended for clinical use if it has the highest level of benefit across a range of clinically reasonable thresholds [19]. We examined the relative benefit of treating patients with FTD/TPI who did not die within 12 weeks against the harms of treating patients who died within 12 weeks. Our study is reported following the TRIPOD guidelines for prognostic model validation [8].

#### 2.2.4. Additional Prognostic Value of Baseline QoL

In the QUALITAS study population, baseline (i.e., at FTD/TPI initiation) overall QoL was measured with the QoL-SS, which encompasses all functional and symptom scales of the EORTC-QLQ-C30 except for financial difficulties [21]. In an exploratory analysis, we re-estimated the logistic regression coefficients before and after adding QoL-SS to the Colon Life nomogram, in order to assess the additional prognostic value of QoL-SS. The additional prognostic value of QoL-SS to the Colon Life variables was assessed by (1) a likelihood ratio test comparing model fit of the Colon Life nomogram with re-estimated coefficients to the ‘new’ nomogram (including QoL-SS), and by (2) comparing performance in terms of discrimination (C-statistic).

### 2.3. Prognostic Value of TK1 Expression

The association between TK1 expression in tumour tissue and survival (PFS and OS) was examined. For QUALITAS patients who consented to using archival tissue for research, formalin-fixed, paraffin-embedded (FFPE) tumour tissue specimens, either biopsies or surgical specimens, were requested via PALGA, the Dutch nationwide pathology archive for histo- and cytopathology [22]. If available, a primary tumour sample and the most recent metastatic sample prior to starting FTD/TPI were requested for each patient. Immunohistochemical staining of TK1 was performed by the University Medical Center Utrecht pathology laboratory using the BenchMark ULTRA fully automated slide staining system (Roch, Rotkreuz, Switzerland). Antigen retrieval was performed with EDTA at pH 9 for 24 min at 100 °C, prior to incubating the slides with antibody (anti-human TK1 rabbit monoclonal antibody, Abcam, Cambridge, UK, ab239885, 1:200 dilution) for 32 min at room temperature. After staining, digital images were captured using the Nano-zoomer XR scanner (Hamamatsu, Hamamatsu City, Japan). TK1 expression was quantified using QuPath v0.2.3 (QuPath, Edinburgh, Scotland) [23]. To automate TK1 expression level analysis, quantification algorithms were generated and repetitively improved until each command was optimized. Selected commands included performing positive cell detection (Appendix A) and tumour versus non-tumour tissue classification. Intensity of TK1 expression was categorized by degree of staining, as 1+, 2+, 3+ for weak, moderate, and strong, respectively. An object classifier was made to classify tumour and non-tumour cells and optimized by assessing its performance in distinguishing tumour and non-tumour cells. In case of uncertainty between tumour and non-tumour parts, the image was assessed by a pathologist. Using the QuPath-obtained values, TK1 expression levels were quantified using two parameters. First, they were quantified as the percentage of tumour cells with a moderate or strong degree (+2 or +3) of TK1 staining, consistent with previously published methods [11] and referred to as the ‘percentage TK1-expressing tumour cells’. Secondly, they were quantified as the H-score (multiplying the percentage of cells in each staining intensity category by the number associated with the category (0, 1, 2, or 3), and summing the results). The H-score results range from 0 to 300, where 0 indicates that ‘all cells are negative’ and 300 indicates ‘all cells are strongly positive’ [24]. Differences in survival in patients categorized per TK1 expression level were compared using log-rank tests in groups based on exploratory cut-off values. A range of cut-off values for the percentage of TK1-expressing cells were selected to ensure that it contained the median value, conform previous studies [11], and spanned the interquartile range values.

### 2.4. Statistical Analysis

*p* values < 0.05 were considered statistically significant, and all tests were two-sided. Analyses were carried out using SPSS version 26.0 and R version 4.0.3. (packages mice, rms, pROC, survival) [25].

## 3. Results

### 3.1. External Validation Cohort

Baseline characteristics of the QUALITAS validation cohort compared with the development cohort and previous external validation cohorts are presented in Table 1. The observed 12-week mortality event rate in the development cohort (30.2%) was considerably higher than our observed 12-week mortality rate (16.7%, 25/150). All patients in the QUALITAS validation cohort and only 6.6% of the patients in the development cohort were treated with FTD/TPI.

### 3.2. Assessment of Colon Life Nomogram Performance in the QUALITAS Dataset

In our cohort, the C-statistic, which assesses discriminative ability, was 0.63 (95% C.I. 0.56–0.70) (Figure 1). The O/E-ratio was 0.52 (95% C.I. 0.37–0.73). The calibration plot indicated systematic evident overestimation of the observed mortality (Figure 2). For comparison, the calibration plot for our cohort is shown alongside the calibration plots for the published external validation cohorts (Appendix A). The calibration intercept was −1.06 (95% C.I. −1.53 to −0.58) and the slope was 0.41 (95% C.I. 0.01–0.81). Figure 3 shows the decision curves for informing clinical decision making regarding FTD/TPI treatment using 12-week mortality risk estimated by the Colon Life nomogram. The net benefit for patients is determined by comparing across three scenarios: non-informed decision making, by (1) treating all patients or (2) treating no patients, and (3) informed decision making, by treating patients according to the nomogram’s predicted 12-week mortality for a given range of thresholds. In the third scenario, patients with a predicted 12-week mortality below the threshold are treated with FTD/TPI, while patients with a predicted 12-week mortality above the threshold are not. The overall benefit is the sum of the net benefit for the treated and untreated patients. Colon Life nomogram guided FTD/TPI treatment, compared to non-informed decision making by treating all or no patients, resulted in a consistently lower overall net benefit for clinically meaningful thresholds (predicted 12-week mortality risk thresholds above 19%). Using nomogram-informed risk to guide clinical decision making for patients who did not receive FTD/TPI treatment, resulted in a net benefit < 0 (harm) for clinically relevant thresholds.

### 3.3. Additional Prognostic Value of Baseline QoL

A low QoL-SS at start of treatment (baseline) was previously found to be independently prognostic for shorter survival in QUALITAS patients [3]. We examined the additional prognostic value of adding baseline QoL-SS to the Colon Life nomogram. After adding baseline QoL-SS to the logistic regression model, the C-statistic improved from 0.69 (95% C.I. 0.62–0.75) for a Colon Life nomogram with re-estimated coefficients to 0.85 (95% C.I. 0.81–0.89; Appendix A). The likelihood ratio test demonstrated a statistically significant improvement in model fit (*p* < 0.005). The distribution of the predicted 12-week mortality is displayed along with the calibration plot in Appendix A.

### 3.4. Prognostic Value of TK1 Expression

We examined the prognostic value of TK1 since high TK1 expression was previously found to be associated with improved PFS and OS in FTD/TPI-treated patients [10,11]. TK1 expression was analysed in archival tissue samples of 110 patients with 146 tissue samples, including 93 primary tumour, 51 metastatic, and 2 recurrence tissue samples (Appendix A). In 36 patients with paired tissue samples, the TK1 expression levels varied substantially between primary versus metastatic tissue (paired sample *t*-test, *p* < 0.05 for H-score and % TK1-expressing cells, Appendix A). In all patients, the TK1 expression levels were significantly higher in primary tumour tissue samples compared to recurrent or metastatic tissue samples (Kruskal–Wallis test, *p* < 0.05, Appendix A).

A significant difference in the median PFS was seen for patients across several TK1 cut-off values (Appendix A). The median PFS was significantly longer in patients with <2.5% TK1-expressing tumour cells, with a median PFS of 104 days (95% C.I. 86–158, *n =* 51) versus 84 days (95% C.I. 72–98, *n =* 100, *p <* 0.05) in patients with ≥2.5% TK1-expressing tumour cells The results remained significant when analysing only primary tissue samples. A similar trend was seen when analysing only metastatic tissue samples, although the result was significant at a higher TK1 cut-off value (15% for percentage TK1-expressing tumour cells). For OS, no prognostic value was seen for TK1 expression level (Appendix A).

## 4. Discussion

The Colon Life nomogram showed evident miscalibration in our real-world population of mCRC patients who were treated with FTD/TPI. Observed and predicted 12-week mortality did not concur. Using the Colon Life nomogram to aid in deciding whether to start FTD/TPI treatment in daily practice would do more harm than good. Guiding clinical decision making with the nomogram resulted in a lower net benefit for clinically relevant 12-week mortality thresholds compared to non-informed decision making through treating all patients. Although the nomogram demonstrated a higher net benefit for 12-week mortality thresholds below 19%, this is not likely to be clinically relevant, since patients will not be denied FTD/TPI treatment based on at most a 19% predicted risk of 12-week mortality. The miscalibration of the Colon Life nomogram in our validation cohort is consistent with previous external validation studies in patients receiving FTD/TPI treatment [6,7]. Yet, the miscalibrated nomogram was deemed a reliable and accurate prognostic model based on its discriminative ability in earlier studies [5,6,7]. This is likely due to the general focus in prediction model studies on the discriminative ability, ignoring the importance of adequate calibration [9,26,27]. Miscalibration can be the result of methodological issues during model development, such as overfitting, or due to differences between the development and external validation cohorts, including patient characteristics, mortality rate, or differences resulting from advancements in treatment for refractory mCRC over time [27]. Without recalibration and further external validation of the updated model, the Colon Life nomogram cannot be used in daily practice to guide decision making for patients who are eligible for FTD/TPI.

In an era in which newly developed prognostic models are increasingly being published each year, external validation and clinical impact studies of existing models are needed to ensure that the models can be used in daily practice [9]. Strengths of our study include external validation of the Colon Life nomogram in patients treated in daily practice with FTD/TPI, in a different geographical area than the original model cohort and adherence to the TRIPOD guidelines. Despite a small sample size with low number of events, we were able to demonstrate evident miscalibration of the nomogram in our cohort.

Given the limited efficacy of FTD/TPI on a group level, methods to improve patient selection are needed to avoid unnecessary exposure to treatment-related toxicity and increase cost-effectiveness. An accurate tool to predict short-term mortality is useful to decide when to refrain from starting FTD/TPI since chemotherapy use near death is likely not beneficial [28,29]. Moreover, the tool can confirm that treatment may be beneficial for patients who are at low risk of short-term mortality. We therefore recommend conducting a new well-powered study, complying to the TRIPOD guidelines to either update/recalibrate the Colon Life nomogram or develop a new prediction model for refractory mCRC patients. Updating prediction models and developing a new model requires a large sample size [14,30,31]. A sufficient sample size might be realized by using individual patient data (IPD) of patients treated with FTD/TPI. In this new or updated model we recommend adding baseline QoL as a predictor for survival, given its strong and independent association with survival [3,32,33] and the large improvement in discrimination we saw in our exploratory analysis. Baseline QoL-SS likely better reflects patient functioning and has superior prognostic value than physician-reported indicators, such as ECOG PS [32,34]. With increasing use of patient-reported outcome measures (PROMs), a prognostic tool which incorporates baseline QoL-SS can be feasible in daily practice. Ideally, this new model will be able to predict mortality in refractory mCRC patients, regardless of the treatment type. However, due to differences in patient populations and treatment effect, models may differ for standard of care treatments versus experimental treatments (e.g., phase I studies).

TK1 expression was identified in the previous literature as a prognostic biomarker for survival in FTD/TPI-treated patients, although results were conflicting [10,11]. Kuboki et al. [10] reported improved OS and PFS for patients with high TK1 expression. Yoshino et al. [11] found that OS was worse in patients with high TK1 expression who received placebo, whereas patients with high TK1 expression showed an improvement in OS when treated with FTD/TPI. These discordant results might be explained by the difference in cut-off values and method of quantification between studies. Moreover, the dual role of TK1 as a predictive and prognostic biomarker complicates the expected effect. TK1 is upregulated during the cell cycle [35], thus a high TK1 expression level may reflect a poor prognosis in mCRC patients. Yet, the incorporation of trifluridine in the nucleus is dependent on TK1 activity [12], so a high TK1 expression may also be predictive for a better response to FTD/TPI treatment. Our results show that TK1 expression levels vary per tissue type, with substantially lower TK1 expression level in metastatic tissue samples than primary samples. A high TK1 expression level was associated with poor PFS, with differing cut-off values for primary versus metastatic tissue samples. In univariable analysis, TK1 expression levels were not associated with OS. The feasibility of TK1 as a biomarker is limited considering the conflicting results and varying cut-off values per tissues sample type, rather than having a consistent accuracy in available archival tissue. Given the differing expression levels, heterogeneous tissue sample types and low number of metastatic tissue samples, we were unable to examine the prognostic value of TK1 in multivariable analysis. Ideally, the prognostic value of TK1 as a biomarker in FTD/TPI-treated patients will be confirmed per tissue sample type.

In conclusion, our external validation study showed evident miscalibration of the Colon Life nomogram, which hampers its use in clinical practice. There is an unmet need for an accurate tool predicting short-term mortality in patients eligible for FTD/TPI treatment, which may be realized by conducting an IPD study with sufficient sample size to enable robust model development to update the Colon Life nomogram or to develop a new model including baseline QoL as a predictor.

## Figures and Tables

**Figure 1 cancers-14-05094-f001:**
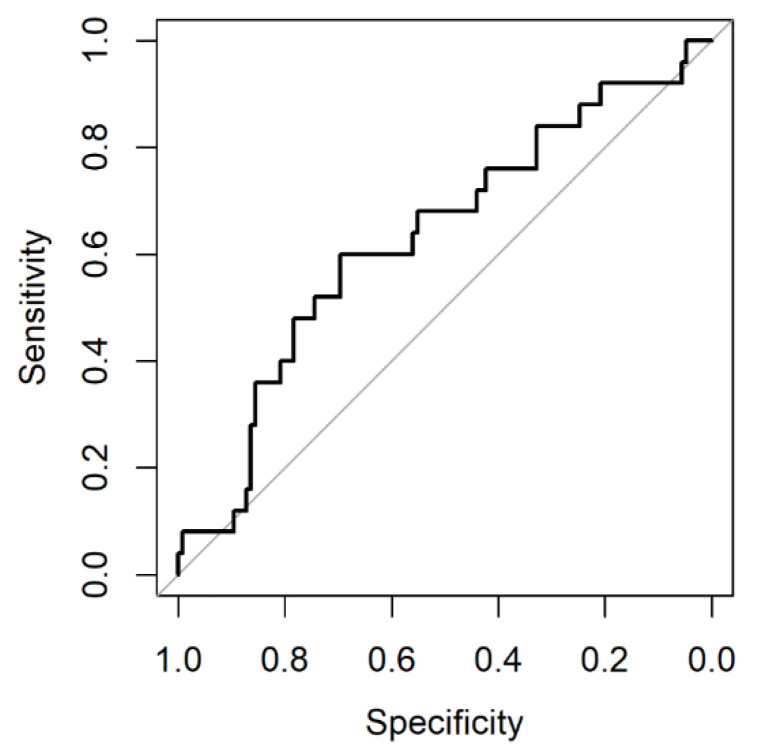
Receiver operator curve for Colon Life nomogram in QUALITAS patients.

**Figure 2 cancers-14-05094-f002:**
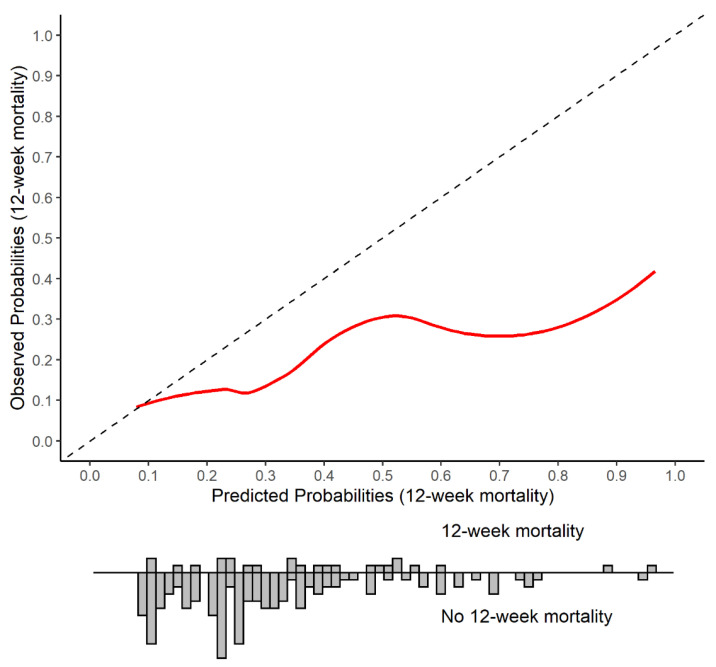
Calibration plot of the Colon Life nomogram: predicted versus observed 12-week mortality in QUALITAS patients. The Colon Life nomogram predicted 12-week mortality versus the observed 12-week mortality is shown, with the diagonal line indicating performance of a well-calibrated model. The histogram displays the predicted probability distribution for patients who died within 12 weeks of initiating FTD/TPI treatment versus who survived.

**Figure 3 cancers-14-05094-f003:**
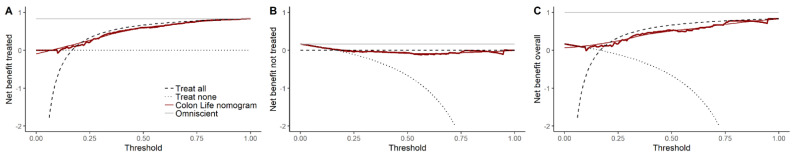
Decision curve analysis indicating the net benefit when using Colon Life nomogram-informed decision making for FTD/TPI treatment. Decision curves based on the net benefit for the treated (**A**), the net benefit for the untreated (**B**), and the overall net benefit (**C**) are shown for informing clinical decision-making regarding FTD/TPI treatment using predicted 12-week mortality risk estimated by the Colon Life nomogram. The net benefit for patients is compared across 3 scenario’s: non-informed decision-making, by treating all patients (“treat all”) or treating no patients (“treat none”), and informed decision-making, by treating patients according to the models’ predicted 12-week mortality risk for a given range of thresholds (“Colon Life nomogram”, thick red line is the Loess smoothed line and the thin red line is the non-smoothed line). The “Omniscient” line indicates the line for a hypothetical all-knowing model. The overall benefit is the sum of the net benefit for the treated and untreated patients. The thresholds shown represent values for 12-week mortality rate. Patients receiving FTD/TPI treatment benefit if the predicted 12-week survival probability is above the threshold value and the patient did not die within 12 weeks (versus harm if they did). Similarly the net benefit for not receiving FTD/TPI is analyzed, where patients benefit if the predicted 12-week survival probability is below the threshold value and the patient did die within 12 weeks (versus harm if they did not). Abbreviations: FTD/TPI (trifluridine/tipiracil).

**Table 1 cancers-14-05094-t001:** Characteristics of the QUALITAS study patients compared to development and external validation cohort patients.

	QUALITAS [3]	Development Dataset [5]	External Validation Dataset [5]	Italian CUP [6]	RECOURSE [7]
**Total study population**	150 (100%)	411	410	341	161 FTD/TPI
**Number of deaths within 12 weeks**	25 (17%)	124 (30%)	89 (22%)	60 (18%)	22 (14%)
**Inclusion period**	2017–2019	2006–2015	2010–2016	2015–2016	2012–2013
**Country**	the Netherlands	Italy	Italy	Italy	Japan, USA, EU, Australia
**Male sex**	102 (68%)	242 (59%)	251 (61%)	212 (62%)	109 (68%)
**Age**					
Mean (±SD)	65.0 (±9.1)	NR	NR	NR	NR
Median (IQR)	65 (59–72)	66 (58–72)	65 (55–71)	61 (NR)	NR
Range	33–86	NR	NR	33–81	NR
<65	68 (45%)	NR	NR	NR	96 (60%)
65–74	60 (40%)	NR	NR	NR	7 (4%)
≥75	22 (15%)	NR	NR	NR	58 (36%)
**ECOG Performance Status**					
0	34 (23%)	194 (47%)	210 (51%)	200 (59%)	87 (54%)
1	63 (42%)	154 (38%)	200 (49%)	132 (39%)	74 (46%)
2	14 (9%)	63 (15%)	0 (0%)	9 (2%)	0 (0%)
Missing	39 (26%)	0 (0%)	0 (0%)	0 (0%)	0 (0%)
**Primary tumour site**					
Right-sided colon	44 (29%)	128 (31%)	135 (33%)	99 (29%)	NR
Left-sided colon	54 (36%)	156 (38%)	179 (44%)	NR	NR
Rectal	52 (35%)	127 (31%)	96 (23%)	NR	68 (42%)
**Primary tumour resection**	113 (75%)	348 (85%)	358 (87%)	312 (91%)	124 (77%)
Synchronous mCRC ^a^	98 (65%)	292 (71%)	272 (66%)	209 (61%)	NR
Metachronous mCRC	52 (35%)	119 (29%)	138 (34%)	132 (39%)	NR
**Molecular pathology** ^b^					
*BRAF* mutation	4 (3%)	13 (3%)	17 (4%)	17 (5%)	NR
*BRAF* wildtype	108 (72%)	219 (53%)	224 (55%)	NR	NR
*BRAF* status unavailable	38 (25%)	179 (44%)	169 (41%)	NR	NR
*RAS* mutation	67 (45%)	167 (41%)	198 (49%)	200 (59%)	NR
*RAS* wildtype	49 (33%)	173 (42%)	133 (32%)	NR	NR
*RAS* status unavailable	34 (23%)	71 (17%)	79 (19%)	NR	NR
MSI	2 (1%)	NR	NR	6 (2%)	NR
MSS	87 (58%)	NR	NR	172 (50%	NR
MS status unavailable	61 (41%)	NR	NR	163 (48%)	NR
**Number of metastatic sites**					
No distant metastasis	1 (0.7%)	0 (0%)	0 (0%)	0 (0%)	NR
1 organ	20 (13%)	87 (21%)	81 (20%)	14 (4%)	NR
2 organs	57 (38%)	172 (42%)	147 (36%)	64 (19%)	NR
3 organs	48 (32%)	NR	NR	122 (36%)	NR
≥3 organs	72 (48%)	152 (37%)	182 (44%)	263 (77%)	NR
≥4 organs	24 (16%)	NR	NR	141 (41%)	NR
**Localization of metastases**					
Liver	115 (77%)	313 (76%)	308 (75%)	267 (78%)	NR
Liver only	10 (7%)	NR	NR	12 (4%)	NR
Lung	102 (68%)	258 (63%)	254 (62%)	255 (75%)	NR
Lung only	6 (4%)	NR	NR	NR	NR
Peritoneal	31 (21%)	95 (23%)	102 (25%)	82 (24%)	30 (19%)
Peritoneal only	4 (3%)	NR	NR	NR	NR
Bone	28 (19%)	36 (9%)	29 (7%)	NR	NR
Brain	3 (2%)	10 (2%)	11 (3%)	10 (3%)	NR
**No. of prior systemic treatment regimens**					
Median (range)	NR	3 (1–7)	3 (1–9)	NR	NR
0	1 (1%)	0 (0%)	0 (0%)	0 (0%)	NR
1	18 (12%)	NR	NR	21 (6%)	NR
2	76 (51%)	NR	NR	93 (27%)	NR
3	41 (27%)	NR	NR	96 (28%)	NR
4	14 (9%)	NR	NR	78 (23%)	NR
5	0 (0%)	NR	NR	31 (9%)	NR
6	0 (0%)	NR	NR	18 (5%)	NR
≥7	0 (0%)	NR	NR	4 (1%)	NR
**FTD/TPI-treated patients**	150 (100%)	27 (6.6%)	100 (24%)	341 (100%)	161 (100%)
**Regorafenib-treated patients**	0 (0%)	113 (27%)	91 (22%)	NR	NR
**Exposure to prior systemic anticancer agents**					
fluoropyrimidine	150 (100%)	NR	NR	337 (99%)	161 (100%)
irinotecan	82 (55%)	NR	NR	334 (98%)	161 (100%)
oxaliplatin	132 (88%)	NR	NR	312 (91%)	161 (100%)
bevacizumab	95 (63%)	NR	NR	294 (86%)	NR
aflibercept	0 (0%)	NR	NR	31 (9%)	NR
anti-EGFR	47 (31%)	NR	NR	143 (42%)	NR
regorafenib	0 (0%)	NR	NR	121 (35%)	NR
**Exposure to treatments:**					
Standard chemotherapy agents ^c^	75 (50%)	NR	NR	NR	161 (100%)
Standard chemotherapy agents ^c^ + bevacizumab	55 (37%)	NR	NR	NR	NR
**Time from diagnosis mCRC to start FTD/TPI (months)**					
Median (IQR)	26.2 (16.8–40.8)	19 (13–29) ^d^	26 (17–40) ^d^	NR	NR
<18 months	43 (29%)	NR	NR	90 (26%)	NR
≥18 months	107 (71%)	NR	NR	248 (73%)	NR
**CEA (ng/mL)**					
Median (IQR)	46 (16–259)	42 (7–190)	58 (15–252)	NR	NR
Mean (SD)	427	NR	NR	NR	NR
Missing	10 (7%)	NR	NR	NR	NR
**Leucocytes (/μL)**					
<10.000	122 (81%)	345 (84%)	336 (82%)	NR	NR
≥10.000	21 (14%)	66 (16%)	74 (18%)	NR	NR
Missing	7 (5%)	0 (0%)	0 (0%)	NR	NR
**Lactate dehydrogenase (U/L)**					
Median (IQR)	272 (210–391)	271 (191–480)	353 (215–529)	NR	351 (245–561)
Missing	7 (5%)	NR	NR	NR	NR

The baseline characteristics of the external validation cohort (QUALITAS study patients) is shown alongside the published characteristics of the development and previously published external validation cohorts. Age, performance status, number of metastatic sites, localization of metastases, number of prior treatment regimens, exposure to prior systemic anticancer agents, CEA, leukocytes and LDH were all determined at refractory disease, prior to starting FTD/TPI treatment. Abbreviations: CRC (colorectal cancer), CUP (Compassionate Use Program), FTD/TPI (trifluridine/tipiracil), IQR (interquartile range), mCRC (metastatic colorectal cancer), No. (number), NR (not reported), SD (standard deviation). ^a^ Synchronous disease was defined as stage IV CRC at diagnosis. ^b^ We assumed that RAS and BRAF mutations are mutually exclusive. ^c^ Prior exposure to fluoropyrimidine, oxaliplatin, and irinotecan. ^d^ Time to chemorefractoriness.

## Data Availability

The data that support the findings of this study are available from the NCR and PLCRC. Restrictions apply to the availability of these data, which were used under license for this study.

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
