# Peer review of "External Validation of the Colon Life Nomogram for Predicting 12-Week Mortality in Dutch Metastatic Colorectal Cancer Patients Treated with Trifluridine/Tipiracil in Daily Practice"

_cancers, 2022, doi:10.3390/cancers14205094_

Round 1
Reviewer 1 Report
Dear Authors
This study indicates the importance of adding QoL (quality of life) to “the Colon Life nomogram” when predicting 12-week mortality of patients with metastatic CRC on FTD/TPI treatment. Though QoL and PS (performance status) which seems similar, the former is one of the patient-reported outcome measures (PROMs) and the latter is one of physician reported indicators. The idea that the PROMs matter is plausible. The conclusion of this study is helpful to readers of this article. However, I’m afraid there are some points should be clarified or revised.
First, I could not find how QoL-SS was integrated into “the Colon Life nomogram”. The nomogram was not shown in the manuscript. Readers of this article cannot calculate 12-weeks mortality with authors’ method even if they got QoL-SS of their own patients. Would you show the new nomogram or succinctly how to calculate it for readers?
Second, multiple comparisons that authors used to discover the optimal thresholds for TK1 and H-score made me dubious about the analysis. I understand that p=0.05 means “1 out of 20 analyses” can be false-positive. Though I’m not a statistician, 30 comparisons (6 (TK1) and 4 (H-score) cutoff points for 3 specimen groups in the supplementary table 2) for both PFS and OS, a total of 60 comparisons seems too many to me. Please make the analysis more succinct.
Some rows in the table 1 slipped off downwards, which make the table confusing (for example, rows of CEA and Leucocytes, the 3rd and the 4th columns). Would you please fix them?
Regards,
Reviewer 2 Report
Ta externally validate existing methods and treatments is an important and necessary part of research and product development. The authors of this article have made a great effort to validate The colon Life nomogram and at the same time evaluate the use of TK 1 as a predictor of outcome.
The methods, the data and and the presentation of the results with analytical discussion is of high quality and I recommend publishing
Author Response
We would like to thank the reviewer for his/her positive comments. We are pleased that the reviewer acknowledges the importance of external validation of prediction models.
Round 2
Reviewer 1 Report
Dear Authors,
Thank you for your revision.
I understand why the authors did not include how the QoL score was included in the nomogram. I’m afraid I am still disappointed with that as one of the readers of this article. The revised abstract clearly shows the meaning of this study compared to the former one.
Table 1 became comprehensible due to the appropriate modification. Thank you.
Regards,